# OpenReview forum: "Graph Attribute Imputation via Generalized Wasserstein Balancing"
_ICLR.cc/2026/Conference — Submitted to ICLR 2026_

### Official Review · Reviewer_VFQw · 2025-10-29

**Soundness:** 2
**Presentation:** 2
**Contribution:** 2
**Rating:** 2
**Confidence:** 4

**Summary:**

This paper addresses the problem of missing node feature imputation in graph-structured data. The authors propose the Relaxed Graph Spectral Discrepancy (RGSD), an optimal transport–based distance that compares graph signal distributions in the spectral domain using the Graph Fourier Transform. To relax the strict mass-matching constraint of standard OT, RGSD employs a flexible mass coupling scheme that mitigates the influence of noisy nodes. Building on RGSD, the paper introduces RGSImp, an iterative alignment-based imputation framework that minimizes RGSD between graph batches to refine missing features. The method requires no masking or parametric models and ensures both statistical consistency and structural preservation.

**Strengths:**

1. The authors present an interesting and well-motivated approach to tackling missing feature imputation in graph-structured data.
2. The paper is clearly written and organized, with a step-by-step explanation that makes the proposed framework easy to follow.
3. The proposed methods appropriately address the key challenges in applying distribution alignment and optimal transport to graph data.

**Weaknesses:**

1. The crucial limitation of this work lies in the assumption that all graphs share an identical structure, even when multiple graphs are present. This assumption substantially restricts the generality and practical applicability of the proposed method. Although experiments are conducted on several datasets, such a setting may rarely occur in real-world applications.

2. The paper lacks an analysis of computational complexity. Both theoretical and empirical runtime analyses should be included to evaluate the scalability of the proposed approach.

3. It is unclear whether diffusion-based imputation methods for tabular datasets [1] or propagation-based imputation methods for graph-structured datasets [2] were considered or whether they are incompatible with the proposed framework. The absence of comparisons with state-of-the-art graph imputation methods raises concerns about the completeness of the evaluation, especially since MICE and MissForest appear as strong baselines.

4. From a presentation perspective, it would be preferable to use bold upright notation for matrices rather than italicized symbols to improve readability and consistency.

[1] Zhang, Hengrui, et al. "Diffputer: Empowering diffusion models for missing data imputation." ICLR 2025.\
[2] Um, Daeho, et al. "Propagate and Inject: Revisiting Propagation-Based Feature Imputation for Graphs with Partially Observed Features." ICML 2025.

**Questions:**

Have the authors considered applying or analyzing different types of missingness? (e.g., node-wise missing)

---

### Official Review · Reviewer_ajXD · 2025-10-29

**Soundness:** 3
**Presentation:** 3
**Contribution:** 3
**Rating:** 6
**Confidence:** 4

**Summary:**

This paper introduces RGSImp, an imputation framework that imputes missing entries in graph node features. RGSImp builds upon the existing alignment-based imputation framework, which assumes that different batches drawn from the same dataset should exhibit consistent distributions. Thus, imputation can be performed by iteratively resampling incomplete batches and updating missing entries to minimize inter-batch distributional discrepancies. When generalizing the alignment-based framework to graph data, the authors observe that existing discrepancy measurements do not consider graph structural information and are sensitive to noise, an issue that is prevalent in graph data, thus degrading imputation performance. To resolve these issues, the authors first propose the Pairwise Graph Spectral Discrepancy (PGSD), a discrepancy measurement designed to compare graphs by capturing both structural patterns and inter-node correlations via the Graph Fourier Transform. Specifically, two graphs are first projected into spectral domains, and then their discrepancy is computed. Then, the authors proposed a new discrepancy matching objective, called Relaxed Graph Spectral Discrepancy (RGSD), which is designed to tolerate noise present in graph data. Finally, the authors introduce the RGSImp imputation framework, which imputes missing values by iteratively reducing the RGSD between sampled graphs. Extensive experiments validate the effectiveness of the proposed RGSImp framework.

**Strengths:**

**(S1)** The paper is clearly written and well structured.

**(S2)** The proposed PGSD measurement computes graph discrepancy by capturing both structural patterns and inter-node correlations. The proposed RGSD objective can tolerate noise present in graph data. The RGSImp imputation framework does not require masking observed entries during training, thereby improving both sample efficiency and usability.

**(S3)** Experimental results on six datasets demonstrate that the proposed RGSImp framework accurately imputes missing values and outperforms classical, alignment-based, and graph-based imputation baselines. Besides, detailed ablation studies verify the effectiveness of the proposed PGSD measurement and the RGSD objective.

**Weaknesses:**

**(W1)** The missingness is simulated under the MCAR setting. The authors are encouraged to evaluate the effectiveness of RGSImp under other missingness mechanisms, such as MAR and MNAR.

**Questions:**

**Q1**. I don't quite understand Figure 1. How should the discrepancy vary with respect to batch size? Should we expect the discrepancy between sampled subsets to naturally decrease as the batch size increases? Do we also expect two sampled subgraphs to become more similar as the batch size grows?

**Minor issues**
1. In Table 2, the MSE results for the MIT-BIH dataset appear to be incorrectly colored.

2. In Line 405, the phrase “presents the average imputation results of KPI and baseline methods” is confusing. It is unclear what KPI refers to in this context.

---

### Official Review · Reviewer_5mac · 2025-10-29

**Soundness:** 2
**Presentation:** 3
**Contribution:** 2
**Rating:** 4
**Confidence:** 1

**Summary:**

This paper addresses the challenge of missing node attribute imputation in graph data (MDFI), a problem that degrades downstream machine learning performance. Existing distribution alignment methods—effective for tabular data—fail on graphs due to two key limitations: (1) standard discrepancy measures ignore structural dependencies between nodes, and (2) noise propagates through graph edges, distorting alignment.

**Strengths:**

1. Diverse datasets, extensive baselines, and targeted ablations (e.g., spectral vs spatial distance) are conducted.
2. This paper addresses the challenge of missing node attribute imputation in graph data (MDFI), a problem that degrades downstream machine learning performance

**Weaknesses:**

1. The paper focuses on noisy node attributes but ignores noisy edges (e.g., incorrect edge presence/absence), which are common in real graphs. It is unclear how edge noise affects RGSD (which relies on Laplacian structure) and whether the method can adapt to it.

**Questions:**

1. Could you add experiments on downstream tasks to validate that RGSImp’s imputations improve downstream performance? This would strengthen the claim that RGSImp solves a practical problem, not just an imputation metric.
2. How does edge noise affect RGSImp’s performance? Could RGSD be modified to handle edge noise without sacrificing structural capture?

---

### Official Review · Reviewer_3JDi · 2025-11-01

**Soundness:** 3
**Presentation:** 3
**Contribution:** 3
**Rating:** 4
**Confidence:** 2

**Summary:**

The paper introduces RGSD, a relaxed graph-spectral discrepancy that compares graph signals in the frequency domain (via GFT) to capture inter-node correlations while down-weighting noisy or missing attributes. Building on RGSD, RGSImp iteratively updates only missing entries by minimizing inter-batch discrepancies, avoiding training a large parametric model on incomplete data. RGSD relaxes OT’s strict mass conservation with a flexible coupling, improving robustness to outliers and corrupted nodes.

**Strengths:**

1. By shifting alignment to the graph-spectral domain and relaxing OT’s mass constraints, RGSD preserves cross-node structure while preventing noisy or missing attributes from dominating.
2. RGSImp optimizes only the missing entries using an inter-batch alignment signal, so it avoids masking observed values and does not require training a heavyweight parametric model.
3. Across multiple datasets, including regimes with high missingness, the method consistently delivers empirical improvements.

**Weaknesses:**

1. The paper lacks empirical validation of when and why OT (and RGSD) is preferable to simpler alternatives, and the current results are not sufficient to substantiate the motivation. Please complement the empirical wins with mechanistic evidence that explains why OT/RGSD helps in the claimed regimes, and add ablation studies, removing relaxed coupling, replacing spectral with spatial, and swapping OT with L2/diffusion, to show that the stated motivation causally drives the gains.
2. "no parametric training" is not a strength by itself. Many prior methods are already lightweight and fast. Please include efficiency comparisons against strong baselines to demonstrate any practical advantage.

**Questions:**

Please refer to the concerns outlined in the Weaknesses.

---

### Meta-Review · Area_Chair_Xk1N · 2026-01-06

**Summary:**

### Summary
This paper proposes RGSImp, an iterative graph node feature imputation framework based on a newly introduced Relaxed Graph Spectral Discrepancy (RGSD). RGSD compares graph signals in the spectral domain using the Graph Fourier Transform and relaxes optimal transport’s strict mass conservation to improve robustness to noise and missing attributes. By minimizing RGSD between sampled graph batches, RGSImp updates only missing entries without training a parametric model, and experiments on several datasets show improved imputation accuracy over existing baselines.

### Strengths

- Introduces a novel discrepancy measure (RGSD) that incorporates graph structure via spectral representations and relaxes OT constraints to handle noise.

- Proposes a non-parametric, iterative imputation framework that avoids masking observed values and heavy model training.

- Empirical results across multiple datasets demonstrate consistent improvements in imputation metrics.

- The paper is generally well written and clearly structured, with ablations supporting key design choices.

### Weaknesses

- The motivation for using OT/RGSD over simpler discrepancies is insufficiently validated; mechanistic evidence and stronger ablations are missing.

- Evaluation is limited to MCAR missingness and largely focuses on imputation metrics, with no downstream task validation.

- Important practical aspects are underexplored, including computational complexity, efficiency comparisons, and robustness to edge noise.

- The method assumes identical graph structures across batches/graphs, which substantially limits generality and real-world applicability, and comparisons with recent state-of-the-art graph imputation methods are incomplete.

While the paper presents an interesting spectral OT-based perspective for graph feature imputation and shows promising empirical results, the current evidence does not convincingly justify the proposed design choices over simpler or more recent alternatives. Key limitations in evaluation scope, missingness assumptions, robustness analysis, and practical applicability remain unresolved. As a result, the contribution is not sufficiently substantiated to meet the acceptance bar at this time.

**Reviewer Concerns:**

No rebuttal

**Reviewer Scores:**

No rebuttal

---

### Decision · Program_Chairs · 2026-01-26

Reject